Persisting roadblocks in arthropod monitoring using non-destructive metabarcoding from collection media of passive traps

Sire Lucas lucas.sire@mnhn.fr 1 2
Schmidt Yáñez Paul 3
Bézier Annie 1
Courtial Béatrice 4
Mbedi Susan 5 6
Sparmann Sarah 3 6
Larrieu Laurent 7 8
Rougerie Rodolphe 2
Bouget Christophe 9
Monaghan Michael T. 3 6 10
Herniou Elisabeth A. 1
Lopez-Vaamonde Carlos 1 4
1 Institut de Recherche sur la Biologie de l’Insecte (IRBI), UMR7261 CNRS - Université de Tours , Tours , France
2 Institut de Systématique, Évolution, Biodiversité (ISYEB), UMR7205 Muséum National d’Histoire Naturelle, CNRS, Sorbonne Université, EPHE, Université des Antilles , Paris , France
3 Leibniz Institute of Freshwater Ecology and Inland Fisheries (IGB) , Berlin , Germany
4 INRAE, UR0633 Zoologie forestière , Orléans , France
5 Museum für Naturkunde –Leibniz Insitute for Evolution and Biodiversity Science , Berlin , Germany
6 Berlin Center for Genomics in Biodiversity Research , Berlin , Germany
7 Université de Toulouse, INRAE, UMR DYNAFOR , Castanet-Tolosan , France
8 CRPF Occitanie , Tarbes , France
9 INRAE ’Forest Ecosystems’ Research Unit Domaine des Barres , Nogent-sur-Vernisson , France
10 Institut für Biologie, Freie Universität Berlin , Berlin , Germany
Nunes-da-Fonseca Rodrigo
Electronic publication date: 2023 Oct 10
Publication date: 2023
Volume: 11
Electronic Location ID: e16022
Received 2023 Feb 17; Accepted 2023 Aug 11
Copyright: ©2023 Sire et al.
Copyright year: 2023
Copyright holder: Sire et al.
License: This is an open access article distributed under the terms of the Creative Commons Attribution License, which permits unrestricted use, distribution, reproduction and adaptation in any medium and for any purpose provided that it is properly attributed. For attribution, the original author(s), title, publication source (PeerJ) and either DOI or URL of the article must be cited.
License URL: https://creativecommons.org/licenses/by/4.0/

Keywords: Bulk metabarcoding, COI, Insects, Malaise traps, Preservative ethanol, Window-flight traps

Funding: CLIMTREE “Ecological and Socioeconomic Impacts of Climate-Induced Tree Dieback in Highland Forests” The French National Research Agency (ANR) ANR-15-MASC-002-01 FEDER InfoBioS EX011185 The German Academic Exchange Service (DAAD) 57440917 The national Museum of natural History (MNHN, Paris, France) The French Office for Biodiversity (OFB) The Federal Ministry of Education and research (BMBF) The Deutsche Forschungsgemeinschaft (DFG) MA 7249/1-1 This work is part of the international project CLIMTREE “Ecological and Socioeconomic Impacts of Climate-Induced Tree Dieback in Highland Forests” anchored in the Belmont Forum Call “Mountains as Sentinels of Change”. The French team (Lucas Sire, Annie Bézier, Béatrice Courtial, Laurent Larrieu, Christophe Bouget, Elisabeth A. Herniou, Rodolphe Rougerie and Carlos Lopez-Vaamonde) was funded by the French National Research Agency (ANR) (ANR-15-MASC-002-01). Lucas Sire, Annie Bézier, Elisabeth A. Herniou and Carlos Lopez-Vaamonde were also funded by FEDER InfoBioS (EX011185). Lucas Sire was also supported by the German Academic Exchange Service (DAAD) (Short-Term Grant 57440917) and by a cooperation and funding agreement (BIOSCAN/FrBOL –SJ 471-21) between the national Museum of natural History (MNHN, Paris, France) and the French Office for Biodiversity (OFB). The German team (Paul Schmidt Yáñez, Susan Mbedi, Sarah Sparmann and Michael T. Monaghan) was supported by the Federal Ministry of Education and research (BMBF) (Förderkennzeichen 033W034A). Paul Schmidt Yáñez and Michael T. Monaghan were also funded by the Deutsche Forschungsgemeinschaft (DFG) (MA 7249/1-1). The funders had no role in study design, data collection and analysis, decision to publish, or preparation of the manuscript.

==============================
Background

Broad-scale monitoring of arthropods is often carried out with passive traps (e.g., Malaise traps) that can collect thousands of specimens per sample. The identification of individual specimens requires time and taxonomic expertise, limiting the geographical and temporal scale of research and monitoring studies. DNA metabarcoding of bulk-sample homogenates has been found to be faster, efficient and reliable, but the destruction of samples prevents a posteriori validation of species occurrences and relative abundances. Non-destructive metabarcoding of DNA extracted from collection medium has been applied in a limited number of studies, but further tests of efficiency are required with different trap types and collection media to assess the consistency of the method.

Methods

We quantified the detection rate of arthropod species when applying non-destructive DNA metabarcoding with a short (127-bp) fragment of mitochondrial COI on two combinations of passive traps and collection media: (1) water with monopropylene glycol (H2O–MPG) used in window-flight traps (WFT, 53 in total); (2) ethanol with monopropylene glycol (EtOH–MPG) used in Malaise traps (MT, 27 in total). We then compared our results with those obtained for the same samples using morphological identification (for WFTs) or destructive metabarcoding of bulk homogenate (for MTs). This comparison was applied as part of a larger study of arthropod species richness in silver fir (Abies alba Mill., 1759) stands across a range of climate-induced tree dieback levels and forest management strategies.

Results

Of the 53 H2O-MPG samples from WFTs, 16 produced no metabarcoding results, while the remaining 37 samples yielded 77 arthropod MOTUs in total, of which none matched any of the 343 beetle species morphologically identified from the same traps. Metabarcoding of 26 EtOH–MPG samples from MTs detected more arthropod MOTUs (233) than destructive metabarcoding of homogenate (146 MOTUs, 8 orders), of which 71 were shared MOTUs, though MOTU richness per trap was similar between treatments. While we acknowledge the failure of metabarcoding from WFT-derived collection medium (H2O–MPG), the treatment of EtOH-based Malaise trapping medium remains promising. We conclude however that DNA metabarcoding from collection medium still requires further methodological developments and cannot replace homogenate metabarcoding as an approach for arthropod monitoring. It can be used nonetheless as a complementary treatment when enhancing the detection of soft-bodied arthropods like spiders and Diptera.

Introduction

Species inventories are a crucial part of ecosystem assessments but are often constrained to a limited number of taxa due to the time-consuming sorting and the need for taxonomic expertise, especially when diverse invertebrate groups are considered (Stork, 2018; Leather, 2018; but see Borkent et al. (2018) and Brown et al. (2018) who morphologically inventoried dipterans in tropical rainforest). A major breakthrough has been the development of batch-species identification with genetic markers using metabarcoding techniques (Yu et al., 2012). Indeed, as this approach identifies species through comparison with DNA barcode reference sequences (Ratnasingham & Hebert, 2007), operators are not required to have taxonomic expertise as long as DNA reference libraries are sufficiently comprehensive and curated by experts (Hebert et al., 2003). Despite the incompleteness of DNA reference libraries, metabarcoding has already proven efficient for monitoring arthropod biodiversity (Yu et al., 2012), including their response to environmental disturbances (Barsoum et al., 2019; Wang C et al., 2021; Sire et al., 2022).

One major shortfall of the metabarcoding approach is the use of destructive DNA extraction from tissue-homogenate after organisms are dried and ground to fine powder (Yu et al., 2012; Sire et al., 2022). As such, both the recovery of abundance data and a posteriori verification of the specimens to confirm the presence of a species in a sample are greatly impeded, if not impossible. Destructive extraction also prevents further study of the material, such as for integrative taxonomic revisions or even new species descriptions (Marquina et al., 2019; Martins et al., 2019). To solve voucher destruction-derived issues, alternative sample preparations have been suggested to facilitate a posteriori morphological control, such as the removal of legs (Braukmann et al., 2019) but is time-consuming, or photographing bulk specimens, which is a more scalable process but may be insufficient for accurate morphological identification.

Insect samplings often require a recipient filled with liquid (e.g., salted water, ethanol, glycol, etc.) to collect them, or an immersion step for preservation, both providing a cheap yet devalued medium from which DNA could be alternatively extracted from. In that sense, Shokralla, Singer & Hajibabaei (2010) successfully sequenced the DNA of insects from the preservative ethanol (EtOH) solution in which they had been stored (both 40% alcohol mezcal and 95% EtOH preservative solutions). A separate study concluded that DNA metabarcoding of preservative EtOH was a reliable way to identify complex freshwater macroinvertebrate samples (Hajibabaei et al., 2012). However, several studies that tried to DNA barcode individual specimens from preservative EtOH reported low amplification success (Robertson et al., 2013; Nassuth et al., 2014). On the other hand, a study of freshwater arthropod communities using metagenomics of preservative EtOH showed accurate and reliable results, though different from those obtained with shotgun-sequencing of pre-sorted morphospecies of the same samples (Linard et al., 2016). In total, we know of 16 other studies that have applied EtOH-based DNA metabarcoding techniques to aim at characterizing complex communities (Zizka et al., 2018; Barbato et al., 2019; Erdozain et al., 2019; Marquina et al., 2019; Gauthier et al. 2019; Martins et al., 2019; Martins et al., 2020; Zenker, Specht & Fonseca, 2020; Couton et al., 2021; Milián-Garcıotaá et al., 2021a; Persaud, Cottenie & Gleason, 2021; Wang Y et al., 2021; Young et al., 2021; Chimeno et al., 2022a; Kirse et al., 2022). Most of these studies found dissimilar communities between EtOH-based metabarcoding and their morphological sorting, bulk homogenate or environmental DNA (eDNA) metabarcoding counterparts and highlighted many technical steps to account for those differences. However cross-study comparisons remain difficult as protocols vary in terms of medium from which DNA is extracted, body structure and size of organisms, primer specificity, bioinformatic pipelines, time prior processing and extraction method (Martins et al., 2020). Along with EtOH metabarcoding, there is a growing interest in the applicability of this method on monopropylene glycol (MPG) solutions. Indeed, MPG is widely used for passive traps as it does not attract insects (Bouget et al., 2009), is cheaper than EtOH, and evaporates less while preserving specimens and their DNA. However, free-DNA preservation in the medium is unknown. So far, direct applicability of metabarcoding approach on MPG-based collection medium has never been tested, and questions remain on the robustness of metabarcoding EtOH- and MPG-based collection media for monitoring terrestrial ecosystems, as very few methodological studies focused on terrestrial arthropods (Marquina et al., 2019; Zenker, Specht & Fonseca, 2020; Chimeno et al., 2022a; Kirse et al., 2022).

The present work had three aims: (i) comparing the species detected using non-destructive metabarcoding with those detected using either morphological analysis or destructive bulk homogenate metabarcoding, (ii) testing the collection medium metabarcoding for two distinct setups commonly used for terrestrial invertebrate biomonitoring, and (iii) clarifying the terminology regarding the nature of the medium from which DNA is extracted to facilitate cross-comparability. Finally, we evaluated the impact of forest disturbance levels on arthropod richness to assess the usefulness of non-destructive metabarcoding technique for wide-scale arthropod biodiversity monitoring programs. To do so, we sampled arthropods in silver fir (Abies alba Mill., 1759)-dominated montane forests along a climate-induced dieback gradient with Malaise trap (MT) and window-flight trap (WFT) setups filled with MPG that was combined with ethanol (EtOH–MPG) and water (H2O–MPG), respectively (Fig. 1). Metabarcoding of DNA from the collection medium (see Box 1) for terminology) was then compared with the results of different treatments of the same traps: destructive homogenate metabarcoding for MT samples, and morphological identification of Coleoptera to species level for WFT samples (Fig. 1).

BOX 1 Terminology and sample types in non-destructive metabarcoding: differences between collection medium and preservative ethanol.

The exploratory nature of non-destructive metabarcoding from various liquids makes comparison difficult, especially due to the type of samples used and the aquatic or terrestrial origin of the targeted arthropod communities (Zizka et al., 2018; Erdozain et al., 2019; Marquina et al., 2019; Martins et al., 2019, Martins et al., 2020; Zenker, Specht & Fonseca, 2020; Milián-Garcıota á et al., 2021a; Persaud, Cottenie & Gleason, 2021; Wang et al., 2021b, Young et al., 2021; Chimeno et al., 2022a). In most of these studies, the word used to describe the sample type is “preservative ethanol”. However, sample type and liquid “clarity”, or “dirtiness” as called by Martins et al. (2019), can be quite different according to facultative pre-processing steps, or the arthropod community targeted, and this may significantly alter the information recovered from metabarcoding. Therefore, we propose a terminology that precisely reflects the sample type used (Fig. B-1).

To illustrate our point, terrestrial arthropods and especially insects are often sampled with passive-sampling trapping methods like Malaise traps (MT) or window-flight traps (WFT). Both collect insects directly within a trapping liquid which stays in the field during a variable time period (e.g. one week to one month). This trapping liquid from which insects are filtered out without further processing is what we call “collection medium”, and is the liquid type used by some studies like Marquina et al., (2019), Milián-Garcıota á et al. (2021a), Young et al. (2021), Kirse et al. (2022), Chimeno et al. (2022a) or to another extent by Swenson et al. (2022) who focused on plants material within Malaise trap samples. Filtered insects can then be morphologically sorted (Young et al., 2021), individually barcoded or processed via metabarcoding from DNA extraction from insects that have been grinded-down to powder (Yu et al., 2012; Sire et al., 2022) that we define here similarly to Marquina et al. (2019) as homogenate metabarcoding. Alternatively, filtered insects can also be placed in fresh ethanol during a variable time period for voucher preservation and storage, and can be filtered out again from this ethanol for further morphological or molecular analyses. The liquid recovered after this second filtration of insects out of ethanol gives a second sample type that we call here “preservative ethanol” and that we consider different from collection medium (Fig. B-1). Currently, this sample type matches the sample description of most of the studies on ethanol-based metabarcoding (Shokralla, Singer & Hajibabaei, 2010; Hajibabaei et al., 2012; Linard et al., 2016; Zizka et al., 2018; Erdozain et al., 2019; Martins et al., 2019; Martins et al., 2020; Zenker, Specht & Fonseca, 2020; Persaud, Cottenie & Gleason, 2021; Wang et al., 2021).

There are notable differences between the two sample types. First whereas preservative ethanol is—as indicated by its name—pure ethanol (which may vary in titrations), collection medium encompasses various chemical compositions based on pure liquids or mixtures (e.g. water, salted water, (monopropylene) glycol, ethanol, ethyl acetate, soap...). Second, collection medium is the dirtiest, as it contains environmental debris and/or arthropod outer-exoskeleton (free-)DNA materials (e.g. pollen, dirt, leave debris, fungi spores, ectoparasites...). Collection medium also contains ingested DNA (iDNA) from intestinal and/or gut contents potentially released by regurgitation and/or defecation death reflexes during insect drowning (Marquina et al., 2019). In comparison, preservative ethanol is relatively clear and free-DNA mostly derives from passive diffusion of the dead arthropods present in the bottle. Of note, the clear/dirty qualification is not binary but rather a continuous gradient that depends of the targeted communities, whether organisms are alive as they get into the liquid used for DNA extraction, or according to the sample’s surrounding environment and its time spent in the field (Fig. B-1). It follows that samples of freshwater communities from the previously listed studies are more similar to preservative ethanol than to collection medium, for three reasons: (i) arthropods are less likely to carry outer-exoskeleton DNA material as evolving in aquatic environments, (ii) after kick-net sampling—that can be extremely dirty—arthropods are often sorted-out of environmental debris prior to ethanol transfer, (iii) life-status prior ethanol transfer is often uncertain (except for live transfer described in Linard et al. (2016)), reducing their potentiality to yield iDNA from similar death reflexes as for terrestrial insects. We acknowledge that these points can be nuanced for kick-net samples (e.g. caddisfly larva cases result in both organic and/or non-organic inputs, kick-net sorting is not compulsory (Pereira-da-Conceicoa et al., 2020), etc) and each case should be explicitly described for further comparisons and robustness.

Information on insect sampling is therefore crucial to correctly categorize the processed samples. Thus, we recommend to distinguish collection medium from preservative ethanol as described above to facilitate cross comparisons between studies and recommend to mention whether arthropods are alive and pre-sorted prior to be transferred in preservative ethanol.

Figure 1 Methodological set-up, traps and treatments processed.

Overview of the trapping methods used in this study. For each type of trap, respective collection media (EtOH–MPG for MT and H20–MPG for WFT) are processed through metabarcoding and compared with different treatments (homogenate metabarcoding for MT and morphological identification for WFT) for species detection. All traps were left one month in the field. Photos credits ©: Malaise trap: Carlos Lopez-Vaamonde; Window-flight trap: Christophe Bouget.

Material & Methods

Arthropod sampling and environmental assessment

Arthropod communities were sampled between May 15th and June 15th of 2017, in 28 silver fir-dominated forest stands in the French Pyrenees, by following two categorical gradients of climate-induced tree dieback and post-disturbance salvage logging (Sire et al., 2022).

In each forest plot, we placed one Malaise trap (MT) in the center, with two window-flight traps (WFTs) facing each other at around 10 m-equidistance from it. All traps were left on-site over the entire mid-May to mid-June period. MT collecting jars were filled with 96% ethanol (EtOH) and monopropylene glycol (MPG) in an 80:20 ratio to limit DNA degradation and EtOH evaporation. WFTs were filled with MPG and water (H2O) in a 50:50 ratio. After one month in the field, sampling bottles were brought back to the lab and stored in a refrigerator at 4 °C for 80–100 days prior to laboratory processing.

Figure B-1 Terminology and description of sample types for metabarcoding from trapping liquids.

Diagram representing the sample types that can be used when metabarcoding collection or preservative media. Solid and dashed violet arrows represent arthropods transferred in and out of liquids, respectively. Arthropod live-status (i.e. dead or alive) and sample condition (i.e. sorted/unsorted) are listed as factors influencing the clarity of the sample. Dotted violet arrows represent arthropod post-processing potentialities (i.e. morphological sorting, DNA barcoding or metabarcoding, storing, etc.). Grey arrows represent time processing that can be variable before sample sequencing. Sample shades of yellow represent the clarity of the liquid sample, with the darker the dirtier according to the gradient of clarity on the right, and with fresh ethanol in light yellow as the clearest and equivalent to a blank control. Sample types boxes are coloured according to the level of sample processing and manipulation post-sampling according to the shaded blue gradient on the right, with light blue the lowest and dark blue the highest amount of sample handling, respectively

Processing of arthropods from MT and WFT samples

We passively filtered the arthropods from the WFT collection media using single-use coffee filters and actively filtered them from the MT collection media using a single-use autoclaved cheesecloth and a Laboport® N 86 KT.18 (KNF Neuberger S.A.S., Village-Neuf, France) mini diaphragm vacuum pump connected to a ceramic-glass filtration column that we decontaminated and autoclaved after each use (see Sire et al., 2022).

Arthropod bulk filtered from collection media were processed differently for each type of trap (Fig. 1). WFTs were used to target saproxylic Coleoptera that, with the help of expert taxonomists, could be morphologically sorted and identified to species level.

As MT recover more diverse and numerous insects, we considered a metabarcoding approach to characterize their derived arthropod communities. Thus, we grounded the collected insects to a fine powder using BMT-50-S-M gamma sterile tubes with 10 steel beads (IKA®; Werke GmbH & Co KG, Staufen im Breisgau–Germany), powered at max speed on an IKA® ULTRA-TURRAX® Tube Drive disperser (IKA®; Werke GmbH & Co KG). We performed DNA extraction from 25 mg (±2 mg) of the arthropod powder using Qiagen Dneasy® Blood & Tissue extraction kit (Qiagen, Hilden, Germany) with final elution in 80 µL of AE buffer (full protocol available in Sire et al., 2022).

Processing of collection media from MT and WFT samples

The collection medium, as opposed to preservative ethanol in various studies, was used as a DNA source in our study (see Box 1). Collection medium processing was performed on 27 MT (one sample was reported missing) and 53 WFT samples (three samples had technical issues in the field). We agitated the sample bottles by hand to ensure the well-mixing of the collection medium and filtered it by pipetting 100 mL with a single-use DNA-free syringe through a single-use 0.45 µm pore size and 25 mm Ø mixed-cellulose ester (MCE) Whatman® filter (Cytiva Europe GmbH, Freiburg im Breisgau, Germany). The filter was held by a 25 mm Ø Swinnex Filter Holder (Merck MgaA, Darmstadt, Germany) that we bleached and autoclaved after each sample filtration. We then placed the filters in DNA-free Petri dishes and left them to dry overnight. After filtering all samples, the filtration step was performed once more with molecular grade water to serve as an extraction blank control.

We extracted the DNA from the dried filters by using the NucleoSpin™ Forensic Filter kit (Macherey-Nagel GmbH & Co .KG, Düren, Germany). The filter was folded and incubated in 600 µL of lysis buffer T1 at 56 °C for two hours with tube horizontally agitated and then centrifugated 1 min 30 s at 11,000 to separate the flow-through from the filter. We then processed this flow-through lysate to carry out DNA extractions on an epMotion® 5075vt (Eppendorf, Hamburg, Germany). We chose magnetic beads to perform DNA extraction and thus used the Macherey-Nagel™ NucleoMag®Tissue kit. We adjusted volumes on the first binding step to the starting volume of lysis buffer accordingly, with 880 µL binding buffer MB2 and 24 µL 0.25X NucleoMag® B-Beads. Extraction was then performed following the manufacturer’s protocol. However, we did the final elution in 100 µL of elution buffer pre-heated at 56 °C with 10 min incubation on beads prior to magnetic separation in an attempt to increase DNA yield. Finally, we quantified each DNA extraction using a Qubit® 2.0 fluorometer and the dsDNA High Sensitivity kit (Invitrogen, Waltham, MA, USA), but few noticeable DNA concentrations were recovered.

PCR amplification for collection media and homogenate metabarcoding

A first but unsuccessful PCR attempt was performed on the DNA extracts obtained from the collection media of both trapping methods to amplify a 313-bp fragment of the cytochrome c oxidase subunit 1 gene (COI) (See Table SI—313_Hom experiment for more details on the PCR conditions, or Sire et al., 2022). However, PCR amplifications of a shorter amplicon to metabarcode collection media were successfully obtained by targeting a 127-bp-long fragment of the COI using the Uni-Minibar primer couple (Meusnier et al., 2008).

We tagged our Uni-Minibar primers to use them in a twin-tagging approach (i.e., identical forward and reverse tag for a given sample). We selected the seven-bp-tags to remain unique after three sequencing mismatches as recommended by Fadrosh et al. (2014). No tag ended in ‘TT’ or ‘GG’ to avoid the succession of three identical nucleotides and potential polymerase slippage. In addition, we added one- to two-bases heterogeneity spacers to shift the position of the start of the read to increase nucleotide heterogeneity in the run (Fadrosh et al., 2014). We checked the red/green nucleotide balance for Illumina MiSeq technology across all designed tags for increasing nucleotide distinction and sequencing quality (see Table SII for the full list of tagged primers).

Before PCR amplification of all collection medium DNA samples, we performed a qPCR optimization using this 127-bp fragment to investigate potential inhibitions and assess the best DNA template dilution. This qPCR trial was carried out using the Uni-Minibar tagged primer couple #96 (Table SII) on 1/10, 1/20, 1/40, 1/80 and 1/160 serial dilution of DNA templates and blank controls in triplicates, amplifying from touch-up PCR cycling conditions (Table S1—127_Opt experiment), and followed by a final acquisition thermal gradient ranging from 65 to 97 °C.

Table 1 Summary of the MOTUs recovery success for each trapping method and sample type analysis.

Traps and treatments	# samples processed	# samples recovered (%)	Min MOTUs per recovered sample	Max MOTUs
per recovered sample	Mean MOTUs
per recovered sample	Total MOTUs
recovered	
MT
( collection media )	27	26 (96%)	3	46	21.81	233	
MT
( homogenate )	27	10 (37%)	17	50	32.4	146	
WFT
( collection media )	53	37 (70%)	1	47	2.06	77	
WFT
( morphology )	53	53 (100%)	22	82	54.43	389	

Then, the PCR amplifications of collection media samples were run in a 20-µL total reaction volume. As DNA concentrations were often too low to be quantified, we used DNA template by volume and not concentration, and amplified it with identical PCR conditions as for qPCR optimization (Table S1—127_Med). We processed all collection medium samples with six replicate PCR reactions, each with a unique primer twin-tag combination from #1 to #31, and samples were distributed in six 96-well plates that also included nine PCR blanks, one filter extraction control for each collection medium and two positive controls.

Finally, we also performed a 127-bp PCR amplification on homogenate DNA extractions of the same MT samples, previously performed in the study by Sire et al. (2022). As we could successfully quantify DNA extracted from tissue homogenate, we distributed DNA template by concentration and not by volume in the PCR mix, and reduced total number of PCR cycles to hit the early exponential phase of the amplification (Table S1—127_Hom, but see Sire et al., 2022). We performed three PCR replicates per homogenate DNA sample, each with a specific primer twin-tag combination from #1 to #30 (two blanks and one positive control included). As part of the study by Sire et al. (2022), these same homogenate samples had also been processed using Leray/Geller primers (Leray et al., 2013; Geller et al., 2013) targeting a 313-bp fragment of the DNA barcode and their results are also used here for comparison with this different PCR treatment.

Library preparation and sequencing of metabarcoding samples

Successful PCR amplification was checked for 10 randomly selected samples for both homogenate and collection media; PCR amplification successes were controlled by migrating 5 µL of PCR product on 2% agarose gel. Homogenate and collection media metabarcoding library preparations were done independently. PCR products of the collection medium samples were purified using CleanNGS (GC biotech, Waddinxveen, Netherlands) magnetic beads at a ratio of 0.8 µl per 1 µl PCR product. Purified PCR product was quantified on a FLUOstar OPTIMA microplate reader (BMG Labtech, Champigny-sur-Marne, France) with the Quant-iT™ PicoGreen® dsDNA assay kit (Thermo Fisher Scientific, Waltham, MA, USA) following the manufacturer’s protocol. Equimolar pooling of the samples was carried out for each plate. An additional step with magnetic beads (0.9:1) was added to concentrate the pools to a total DNA quantity of 35 ng of purified amplicon in a final volume of 50 µL. For the library preparation of the pools the NEBNext® Ultra™ II DNA Library Prep Kit for Illumina® (New England Biolabs, Ipswich, MA, USA) was used following the manufacturer’s protocol. Adaptors were diluted 10-fold and a clean-up of adaptor-ligated DNA without size selection was performed. The PCR enrichment step used forward and reverse primers that were not already combined and three amplification cycles. Sequencing was done on an Illumina MiSeq platform using V3 2 ×300 cycle kits.

Bioinformatic and statistical analyses

Bioinformatic processing was performed following the DAMe pipeline (Zepeda-Mendoza et al., 2016, as in Sire et al., 2022). A various number of PCR replicates were investigated to retain shared MOTUs with a minimum of two reads in collection medium metabarcoding (i.e., in at least 1/6 PCR replicates, standing as additive combination of sample replicates; or 2/6; 3/6 and 4/6 for relaxed restrictive combinations). For homogenate metabarcoding two PCR replicates (2/3) with two reads minimum per MOTU were retained to discard singletons.

MOTU clustering was performed using a 97% similarity threshold and taxonomic assignment was performed with the BOLD DNA reference database (Ratnasingham & Hebert, 2007) using BOLDigger tool with BOLDigger option (Buchner & Leese, 2020). Therefrom, taxonomy was retained based on the maximum similarity value of the top 20 hits and correction of top hits was then performed based on the BOLD identification API (Buchner & Leese, 2020). MOTUs with identical species-level taxonomic assignment were then merged manually. Comparisons of MOTUs consensus sequences between collection medium and homogenate metabarcoding were performed with BLAST+ (Camacho et al., 2009). A threshold of read number defined by the lowest mean between collection medium and homogenate treatments was applied to consider samples in further analyses. Hence, only samples with >10k reads were retained and considered as samples that could be detecting a representative richness for the given trap types.

All statistical analyses were run with R v4.1.0 (R Core Team, 2017) to test for differences in MOTU recovery between collection medium and homogenate metabarcoding. MT homogenate metabarcoding results of 127-bp amplicons from Uni-Minibar primers were also compared with homogenate metabarcoding of 313-bp amplicons of the same traps (Sire et al., 2022). To do so, we checked for homoscedasticity of variance and normality of data using ‘descdisc’ and ‘fitdist’ functions from the fitdistrplus v1.1-6 package and assessed with Levene test. If data were normally distributed, an anova test was applied, followed when significant by a pairwise T-test with Bonferroni correction using R built-in functions. If non-parametric analyses were needed, Kruskal-Wallis test was applied, along with unpaired Wilcoxon rank-sum test with Bonferroni correction to assess the direction of the significance when needed. Similar analyses were performed to account for the difference in species richness across dieback level gradient and stand types.

Results

Bioinformatic processing and taxonomic assignment

Sequencing all collection media samples (EtOH–MPG and H2O–MPG) resulted in 12,686,324 reads in total (see Table SIII and supplemental information for more details). Processing the window-flight traps (WFTs) using 1–4/6 replicate combination parameters, collection medium sequencing yielded 191, 77, 53, and 37 MOTUs respectively (Table SIV). In all cases, more than half of MOTUs were represented by Diptera, ranging from 52–56%. However, only 4–12% Coleoptera MOTUs were recovered, albeit being the main taxonomic group sampled by WFT (Table SIV). In comparison, morphological sorting of the WFT led to 389 Coleoptera morphotaxa, of which 343 species could be identified (Table V). Out of 20 Coleoptera found with the 1/6 combination parameter, 18 (90%) were identified to the species level. Among these, 12 species were also found in the morphological dataset, of which only five were found by metabarcoding and morphology treatments of the same traps. However, these observations had low reliability since these five species remained undetected by metabarcoding in most of the traps in which they had been identified morphologically, and multiple detections in metabarcoding samples were conversely not verified via morphological sorting (e.g., potential cross-contaminations). Similarly, for the three Coleoptera from 2/6 combination parameter that were all identified down to species level: Cis festivus (Panzer, 1793), Pyrochroa coccinea (Linnaeus, 1761) and Quedius lucidulus (Erichson, 1839). P. coccinea was not found in the morphological dataset and the other two also corresponding to the Coleoptera MOTUs found in 3/6 and 4/6 combination parameters were present but not detected concurrently in the morphological and metabarcoding treatments of the same traps (Tables SV, SVI).

For the Malaise trap (MT) collection media, ratios in MOTU reduction from the various filtering steps were similar for all combination parameters apart from the additive one (1/6 PCR replicates) which showed a more drastic decrease in both reads and MOTUs (Fig. S1, Table SIII). We compared 1/6 and 2/6 combination results to 313-bp bulk metabarcoding results from a previous study on the same MTs (Sire et al., 2022). As the two COI fragments were of different lengths (127 and 313-bp) and did not overlap (Elbrecht et al., 2019), we downloaded full-length barcodes of publicly available records matching identification from BOLD for 313-bp derived MOTUs. Comparisons with our 127-bp derived MOTUs from 1/6 and 2/6 combination parameters gave only 67 (114 with > 97% similarity) and 45 (72 with > 97% similarity) identical and shared MOTUs, respectively. Comparing both 127-bp combination parameters, 40 MOTUs with 100% similarity to 313-bp dataset were shared. The additional 27 MOTUs from the additive combination are identified as Diptera (16), Lepidoptera (six), Hemiptera (two), Coleoptera (two) and Hymenoptera (one).

While 1/6 additive combination allows a slightly better recovery of insects from collection medium metabarcoding of MT samples, no improvement was highlighted under that parameter for WFTs. As this led to little increase in MOTUs number, and in order to reduce the risks of dealing with false positive MOTUs from 1/6 PCRs additive combination, hereafter results focus on the filtered dataset from the 2/6 PCR replicates relaxed restrictive combination only. The 27 EtOH–MPG (MT) samples gave a total of 238 arthropod MOTUs and a number ranging from three to 46 (Table 1) with 147,358.6 (±13,687.25 SE) reads per sample. As one trap had <10k reads, it was further removed, giving a final dataset of 233 arthropod MOTUs for 26 successfully metabarcoded samples. Of the 53 H2O–MPG (WFT) samples, 37 (70%) yielded arthropod MOTUs for a total number of 77 (Table 1; Table SVI), 12,176.06 (±5,073.41 SE) reads per sample, with MOTUs number ranging from one to six for all but one sample that harboured 47 MOTUs and a mean of 2.06 MOTUs per sample (Table 1). Similar percentages of taxonomic assignment were found for the 233 MOTUs detected in the MT collection medium (EtOH–MPG), 118 (51%) were unambiguously assigned to species (Fig. 2A; Table SVII).

Figure 2 Taxonomic assignment of recovered arthropod MOTUs from various metabarcoding treatments (collection medium or homogenate) or primer sets (Uni-Minibar or Leray/Geller).

Number of MOTUs detected from collection medium (yellow) or homogenate metabarcoding (blue) with Uni-Minibar primer set or from homogenate metabarcoding using Leray/Geller primer set (grey) of the same Malaise trap samples and taxonomically assigned unambiguously with a 97% threshold based on BOLD DNA barcode reference libraries. Data for each treatment and primer set are shown (A) for the total MOTU richness, and (B) for their four most diverse arthropod taxa, respectively. The total number of MOTUs and MOTUs identified to species level are displayed with a dark-to-light shaded colour gradient—yellow, blue or grey respective to each treatment—and with labels providing the number of MOTUs.

Sequencing of MT tissue homogenate targeting the 127-bp amplicon resulted in 3,728,546 reads in total, reduced to 406,776 for 169 MOTUs after applying a relaxed restrictive combination parameter of 2/3 PCR replicates. Filtering of negative and positive controls generated 75% reads drop (from 406,776 reads to 101,655 for a three MOTUs loss). Two traps yielded no result with homogenate metabarcoding and corresponded to samples with 29 and 46 MOTUs detected in collection medium. Each of the 25 remaining traps harboured one to 50 MOTUs and an average number of reads per sample of 10,982.3 (± 4,139.802 SE). For ecological analyses, 15 traps did not meet the >10k reads threshold and were discarded, leading to a final dataset for homogenate metabarcoding from MT samples comprising 146 arthropod MOTUs for 10 traps (Table SVIII). Taxonomic assignment resulted in 144 (99%) MOTUs assigned to order and 115 (79%) to species (Fig. 2A). Compared with metabarcoding of the same traps targeting a 313-bp amplicon (Sire et al., 2022) that produced 962 MOTUs of which 539 were identified to species (Fig. 2A), our results for a shorter fragment (127-bp) yielded a significantly lower number of MOTUs per trap overall (Wilcoxon rank sum-test: p = 1.3e−05; Fig. 3), as well as across different taxa (Fig. S2). Overall, dipterans were the most diverse group recovered regardless of the method, with 51%–80% species level identification success according to the treatment and primers used (Fig. 2B). Further analyses of community diversity only focus on the results of the 127-bp homogenate metabarcoding for comparisons with Malaise trap collection medium metabarcoding using that same shorter fragment.

Figure 3 Comparison of MOTU richness recovered from Malaise traps using various metabarcoding treatments (collection medium vs. homogenate) or primer sets (Uni-Minibar vs. Leray/Geller).

Boxplot of MOTU count for collection medium (yellow) or homogenate metabarcoding (blue) with Uni-Minibar primer set or from homogenate metabarcoding using Leray/Geller primer set (grey) of the same Malaise trap samples. Black dots represent samples considered after bioinformatic processing and data curation. Significant differences adjusted with Bonferroni correction are highlighted with ‘*’ and ‘N.S.’ stands as non-significant. Similar MOTU richness could be detected from collection medium and homogenate metabarcoding using Uni-Minibar primers, but significantly lower than the richness detected with a longer amplicon targeted with Leray/Geller primers in a previous study (Wilcoxon rank sum-test: 1–2: p = 0.071; 1–3: p = 1.3e-09; 2–3: p = 1.3e-05).

Comparative analyses of community composition between treatments and across forest disturbances

Metabarcoding analyses of the WFT collection medium samples yielded 77 MOTUs, with only three Coleoptera and no joint observation with morphological treatment of the same samples. Thus, we focus hereafter on the results from MT samples only. Overall, the MOTU richness from collection medium metabarcoding (n = 26, mean = 21.80, median = 20.5) was similar than with homogenate metabarcoding (n = 10, mean = 32.4, median = 31.5) (Wilcoxon rank sum-test: 1–2: p = 0.071; Fig. 3). However, community compositions differed between both treatments. A higher proportion of insect MOTUs was recovered from MT homogenate (94%–137 out of 146 arthropod MOTUs) than from MT collection medium (85%–198 out of 233 arthropod MOTUs), with the remaining being Collembola and Arachnida (Figs. 4A, 4B). Insects recovered from the EtOH–MPG collection medium were mainly represented by Diptera (77%–153 MOTUs) out of 11 insect orders (Fig. 4C). The insect community from homogenate was composed of eight insect orders and a MOTU distribution less biased toward Diptera, representing only 65% (89 MOTUs) of the total 233 MOTUs (Fig. 4D).

Figure 4 Taxonomic composition (number of MOTUs) of arthropod communities recovered from both homogenate and collection medium metabarcoding treatments of Malaise trap samples.

Taxonomic composition (% (italics) and absolute numbers are reported) of MOTUs retrieved from (A, C) collection medium metabarcoding (B, D) and homogenate metabarcoding of the same Malaise trap samples. A & B show the number of MOTUS per Arthropoda classes recovered from homogenate and collection medium respectively. C & D show the four insect orders with the highest number of MOTUs for homogenate and collection medium respectively. Insects included in the “Others” category belong to Neuroptera, Psocodea and Raphidioptera as well as to (C) Ephemeroptera, Mecoptera, Thysanoptera and Trichoptera in collection medium and (D) Hemiptera in homogenate.

The numbers of detected MOTUs for non-insect taxa (e.g., Collembola and Arachnida) was significantly higher in collection medium than in homogenate metabarcoding (Pairwise T-test: 1–2: p = 6.6e−03), similar for Diptera (Wilcoxon rank sum-test: 1–2: p = 0.15) and the category “other insect orders” (W-test: 1–2: p = 1), but significantly lower for Coleoptera (W-test: 1–2: p = 3.9e−03), Hymenoptera (W-test: 1–2: p = 1.9e−03) and Lepidoptera (W-test: 1–2: p = 1.4e−02) (Fig. S2).

Comparisons of MOTU consensus sequences between collection medium and homogenate metabarcoding gave 71 exact MOTU matches (Fig. 5A), of which 18 suggesting that DNA from the same individuals can genuinely be recovered by both treatments of the same sample. When considering MOTUs that were identified to species level—118 out of 233 for collection medium and 115 out of 146 for homogenate metabarcoding (Figs. 2A; 5B)—40 species were shared between both treatments (Fig. 5B). However, only nine species were recovered by both treatments of the same sample (Table SIX).

Figure 5 Taxonomic composition (number of MOTUs) of arthropod communities recovered from both homogenate and collection medium metabarcoding of Malaise trap samples.

Venn diagram of (A) the total number of MOTUs or (B) MOTUs identified to species level for homogenate metabarcoding (blue) and collection medium (yellow) of Malaise trap samples. (A) Seventy-one MOTUs are shared between collection medium and homogenate metabarcoding, while (B) 40 species are shared by both sample types.

We detected no significant change in MOTU richness in collection medium of MT samples among dieback levels (anova: df = 2, p = 0.91) or stand types (anova: df = 2, p = 0.634) (Fig. 6).

Figure 6 Variation in MOTU richness across natural and anthropogenic disturbances.

Comparison of MOTU richness recovered from collection medium metabarcoding. Richness variations are tested across (A) low, medium and high climate-induced dieback levels and (B) between disturbed but unmanaged and salvage-logged plots. Black dots represent samples. No significant differences could be detected with anova tests for both disturbances’ gradients (Dieback level: df = 2, p = 0.91; Stand type: df = 2, p = 0.634).

Discussion

From fieldwork to bioinformatic processing—technical considerations for collection medium metabarcoding

DNA metabarcoding from bulk samples of arthropods has flourished in the past 10 years, and with it arose many technical considerations from the experimental to the bioinformatic processing steps (Alberdi et al., 2018; Elbrecht et al., 2019). A number of limitations have been identified for DNA metabarcoding from collection medium and preservative ethanol (Martins et al., 2020), but studies remain scarce. Our analyses corroborated the possibility to detect species from collection medium with metabarcoding, but the low richness of MOTUs we detected in most samples was clearly not representative of the insect diversity collected in traps. Below, we discuss how some steps in the process may directly impact metabarcoding based on collection-media, and propose further investigations to test and improve the efficiency and robustness of the approach.

Beginning with the field sampling step, one factor that could explain the low number of MOTUs detected in collection-media is the fact that trap jars are often set in clearings or open canopies, where they are exposed to warm temperatures and direct UV-light. Both of these are likely to accelerate DNA degradation in the field. Drowned organisms also passively release water by osmolarity and dilute the collection medium, which might reduce its preservative capacity and increase DNA hydrolysis when biomass accumulates in the trap (Jo et al., 2019). These factors, extended over our one-month period of field sampling and added to the subsequent storage of the samples in a cold room, most likely led to DNA degradation and explain the low MOTUs detection rate. It is advisable to replace the bottles of malaise traps every one-to-two weeks to minimize DNA degradation and optimize passive diffusion (Martins et al., 2019), with sample storage (or pre-processed filters in case of storage shortage) at −20 °C (Yamanaka et al., 2016).

The chemical composition of the collection medium may also play a critical role on the preservation of extracellular free DNA (i.e., DNA molecules passively released by organisms into the collection medium). Water should be minimized to avoid DNA hydrolysis (Jo et al., 2019); however, substitution with ethanol in WFTs can lead to higher evaporation rates and costs, increased attractiveness to some insect groups and subsequent sampling biases (Bouget et al., 2009). Furthermore, WFTs are exposed to rainfall due to their wide opening on the collector and thus prone to increased water content. The volume of rainfall can be reduced by drilling small holes into the container, but this can lead to liquid loss and extracellular DNA dilution. Alternative collection medium for WFT include NaCl-H2O solution which was successfully metabarcoded in two studies (Milián-Garcıota á et al., 2021a; Young et al., 2021), drawing a parallel with eDNA metabarcoding of water samples from marine ecosystems. Salted water has been shown to be cost-effective for monitoring Coleoptera after a 4-week sampling period in the field (Young et al., 2021; ) but may further degrade DNA in traps that collect soft-bodied taxa for which DNA is passively diffused more quickly in the collection medium. Milián-Garcıotaá et al. (2021b) successfully adapted this method to target Diptera, although their report was based on two samples and 42 individuals representing five species, limiting insights for its use for more complex communities. Our attempt to metabarcode insects from MPG-based collection medium (H2O-MPG in a 50:50 ratio) proved not suitable under these sampling conditions. In addition to potential DNA degradation, the high viscosity of MPG (Martoni et al., 2021) might facilitate individual escapes in the trap due to increased floatability (McCravy & Willand, 2007), may coat organisms and free DNA molecules and/or clog the filter membrane (as experienced when filtering 100 mL of collection media containing MPG at 50% only). All of these could reduce the passive diffusion of DNA from organisms and the recovery of DNA during extraction. Martoni et al. (2021) did amplify DNA—but did not confirm insect identification nor ruled out contaminant amplification through sequencing—from MPG-based solution. Additionally, they targeted a single species with controlled abundance, and added lysis buffer with the specimens prior to filter MPG for DNA extraction. Currently, MPG-based metabarcoding of whole communities in controlled conditions is yet to be tested, and while MPG has been shown to be a good preservative of DNA that was subsequently extracted from organisms (Stoeckle et al., 2010; Höfer et al., 2015; Nakamura et al., 2020; Martoni et al., 2021), whether free DNA molecules degrade in MPG-based collection medium remains unknown.

In our study, protocol discrepancies between treatments are likely to have resulted in some of the observed variation. Several steps during the wet-lab processing may have impacted DNA recovery. Both for EtOH- and MPG-based collection medium, how free DNA molecules are impacted by the pre-filtering of the arthropods from the medium is unknown and often overlooked. Coffee filters used for WFT vs. cheese-cloth for MT may have differently retained free-DNA molecules and impeded subsequent DNA extractions. Nevertheless, we believe that the choice of filters that are used for direct DNA isolation from collection media is the most critical filtering step. Capture efficiency depends on DNA polarity which may be affected by the chemical composition of the collection medium. Following Li et al. (2018), we chose mixed-ester cellulose filters for all our collection media samples. Other studies successfully captured DNA with nitrate filters from preservative ethanol (Milián-Garcıotaá et al., 2021a; Milián-Garcıotaá et al., 2021b; Young et al., 2021), with an additional grinding step of the membrane to increase lysis efficiency (Kirse et al., 2022). Furthermore, collection medium might also accumulate inhibitors released from arthropods (Boncristiani et al., 2011; Linard et al., 2016) or from external by-catches (e.g. leaves or pine needles releasing pigments and terpenes (Tang, Zhao & Liyan Ping, 2011), molluscs or worms with high polysaccharide contents), that are likely retained by the filter. Similar inhibition and DNA purity issues have been reported for non-destructive lysis buffer extractions (Kirse et al., 2022). Thus, questions on DNA-binding and polarity, filter capture and retention capacities, or pore size and fluidity/clogging remain and should be further explored to evaluate the impact on both free DNA and potential inhibitors yielded from different EtOH-based solutions (and non-destructive alternatives more generally; Kirse et al., 2022).

The use of different DNA extraction methods may also have impacted the results of our comparisons between our treatment of MT in the present study (we used magnetic beads, following Martins et al. (2019) in an attempt to favour DNA yield) and the one previously used for extracting DNA from homogenates of the same samples (using silica column; see Sire et al. (2022)). Nonetheless, rather reassuringly, the results of our comparisons on variable MOTU compositions across treatments are consistent with previous findings where extraction methods were identical (Marquina et al., 2019).

PCR primer efficiency is also a key factor in metabarcoding (Martoni et al., 2022), and we found lower MOTU richness with Uni-Minibar primers compared to the more commonly used 313-bp COI fragment amplified by the mlCOIintF/jgHCO2198 primer set (Leray et al., 2013; Geller et al., 2013). Unfortunately, our attempts with the latter failed on collection media, most likely because of low DNA concentration and degradation of free DNA in the collection medium due to field and storage conditions. Thus, our comparison between treatments is also a comparison of primer sets, and because the longer amplicons allowed increased resolution (Fig. 3, Fig. S2 ), it is likely that similar amplification and identification biases were obtained from metabarcoding the collection media.

Bioinformatic processing is also instrumental for the determination of MOTU diversity. In particular, processing parameters and the strategy for filtering MOTUs across different PCR replicates can greatly impact the number of sequence reads and MOTUs retained (Alberdi et al., 2018). Regardless of the type of trap (WFT and MT), the use of a more conservative retention (MOTUs present in at least two PCRs) allowed a drastic reduction of unknown sequences and chimeras, untargeted organisms, or contaminants, but did not lead to a notable decrease in identified and plausible species. It also shows that sequencing depths allocated to sequence species present in the samples was too low based on the numbers of samples not considered after applying a 10K reads threshold, further influencing the poor results on our MOTU recovery. More experiments in controlled conditions are required to better grasp and understand all the factors potentially affecting the outcomes at each step of the bioinformatic analysis.

Community analyses and terrestrial insect monitoring from EtOH-based collection medium metabarcoding of MT samples

Accurate species identification is an important part of ecological analyses, and is needed to unravel species biology and the functions they may have in their respective environments (Tautz et al., 2003). In environmental genomics, community analyses based on metabarcoding rely on DNA reference libraries to identify species. While the metabarcoding of collection medium allows for the preservation of voucher specimens for morphological validation, it remains important to assess whether this molecular approach can reliably characterise insect communities. Taxonomic assignment at the species level was lowest for Diptera (51%), Arachnida (16%) and Collembola (10%). This may be explained by the fact that these groups are highly diverse, can be difficult to identify based morphological criteria, and are poorly covered in DNA barcode reference libraries (Morinière et al., 2019; Sire et al., 2022). Thanks to the recent DNA barcoding efforts to cover the fauna of Germany, it is possible to identify a relatively large proportion of the Central and Western European Diptera (Morinière et al., 2019). It is also of note that the short length of the amplicon targeted here (127 bp) reduces taxonomic resolution (Hajibabaei et al., 2006; Meusnier et al., 2008; Elbrecht et al., 2019).

We found that the arthropod communities characterised with collection medium metabarcoding and homogenate metabarcoding for the same MT samples were dissimilar, with only 71 MOTUs / 40 identified species shared between treatments (Fig. 5), with variations arising both at class and order levels, respectively. This community dissimilarity highlighted between collection medium and homogenate metabarcoding treatments of a single MT sample are in line with previous reports.

Both methodological and biological factors may influence this community dissimilarity. First, we recovered few Hymenoptera MOTUs, and conclude that this was caused by a low affinity of Uni-Minibar primers for this order (Yu et al., 2012; Brandon-Mong et al., 2015), and suggests degenerate primers are needed for the analysis of complex communities. To that extent, fwhF1–fwhR1 primer couple designed for freshwater invertebrates by Vamos, Elbrecht & Leese (2017) could represent an efficient alternative to be considered in future research. Second, smaller organisms or tissues from fragile ones (e.g., Collembola or Arachnida) may pass through the filter mesh during the pre-filtering step, or may have been tightly retained within the filter that was placed back in the collection medium prior DNA extraction, allowing them to release more DNA material into the sample (Marquina et al., 2019). Although our results for Lepidoptera were similar across treatments, filtration could provide an explanation for the high detection described by Chimeno et al. (2022a), considering the wings scales detaching and freely floating into the trap or preservative solution. Third, species detection in collection medium has been shown to be inversely proportional to sclerotization. Soft-bodied (poorly sclerotized) arthropods like Arachnida, Collembola and Diptera are often well-represented, while Coleoptera and Hymenoptera are often under-detected (Marquina et al., 2019; Kirse et al., 2022; Chimeno et al., 2022a; Martoni et al., 2022). This also likely explains the failure of MPG-based collection medium metabarcoding from WFTs mainly targeting Coleoptera, that only accounted for 4% of the recovered diversity and were not consistent with morphological treatment, hence preventing the method to reliably reflect their expected diversity. Finally, prey-derived DNA that is regurgitated or defecated by captured organisms at the time of death within the collection medium (Marquina et al., 2019) can be a significant source of variation between treatments, and may explain an overrepresentation of insects known to be prey, such as Lepidoptera (Chimeno et al., 2022a) and perhaps Diptera in our study.

Dipterans are a highly diverse and functionally important group of insects in forest ecosystems (Mlynarek, Grégoire Taillefer & Wheeler, 2018; Chimeno et al., 2022b), and taking advantage of their greater detection in EtOH–MPG collection medium metabarcoding could improve our understanding of their ecological role for environmental assessment. However, collection medium metabarcoding is unlikely to replace homogenate metabarcoding (Marquina et al., 2019). Running both treatments in parallel could instead enrich biodiversity surveys and broaden our understanding of trophic assemblages—but not interactions. In particular, medium-based metabarcoding may outperform bulk-based approaches for the detection of prey DNA from gut content or, in a more exhaustive scale, for the recovery of DNA from fungal spores, pollen or other plant material brought by the arthropods falling in the traps, and for which a recent study could successfully detect red-listed or neophyte plant species in or around protected areas (Swenson et al., 2022). The caveat of homogenate metabarcoding remains the loss of voucher specimens which hinders morphological studies and the DNA barcoding of individuals (Marquina et al., 2019). This may also hinder the transition for metabarcoding-based biodiversity survey if sample preservation is legally mandatory in official biomonitoring programs (Martins et al., 2019). This problem may not apply to other types of samples as in surveys of freshwater organisms, similar taxonomic recoveries were found by metabarcoding EtOH preservative and homogenates (Hajibabaei et al., 2012; Zizka et al., 2018). As there are no standardized laboratory procedures, comparisons between sample types and studies remain difficult. However, these discrepancies in species recovery patterns may reflect the differences among sample types and highlight the need to assess sample provenance and medium clarity for reliable comparisons (Box 1; Martins et al., 2019; Martins et al., 2020).

Although metabarcoding collection medium or homogenate documented different arthropod communities, both methods may allow us to monitor the response of species richness to environmental changes—in our case, the response of arthropod richness to forest dieback induced by drought and associated forest management. We detected no response in terms of MOTU richness across the three levels of climate-induced forest dieback intensity, nor between the three various stand types. This result is similar to a previous broader study that included the samples analysed here (Sire et al., 2022). However, the robustness of this result should be taken with caution as we could not statistically compare the MOTU richness of the homogenate metabarcoding amplified using Uni-Minibar primers due to the low number of traps considered. Additionally, the relatively low success of MOTU recovery from collection medium metabarcoding also prevented further investigations on community structure to compare with changes in species and functional compositions highlighted by Sire et al. (2022). For that matter, Chimeno et al. (2022a) showed that Malaise trap communities across their two treatments (i.e., preservative EtOH vs. homogenate metabarcoding) were dissimilar. They also showed that variation in community composition obtained from EtOH-based metabarcoding was driven by seasonality rather than by ecological gradients. In addition, they suggest that DNA from prey species derived from gut-contents present in the ethanol can bias seasonal patterns. This contradicts previous studies on freshwater ecosystems that highlighted the potential to monitor communities (Hajibabaei et al., 2012; Zizka et al., 2018; Martins et al., 2019; Martins et al., 2020; Persaud, Cottenie & Gleason, 2021) or to unravel genetic structure of populations (Couton et al., 2021) with EtOH-based metabarcoding, depicted as one efficient non-destructive alternative to homogenate metabarcoding. However, DNA and organismal dispersions within water are not analogous with terrestrial collection medium samples, and the gut-derived information blurring the DNA source and organism interactions further impedes the disentanglement of potential ecological signal in the data.

In light of the variability of the results across treatments, other alternatives are being developed to recover comparable species inventories with no voucher destruction, or added information such as species abundance. As such, non-destructive DNA extraction buffer (e.g., a mixture of lysis buffer with chaotropic salts and proteinase K) is considered as an alternative to keep vouchers intact (Carew, Coleman & Hoffmann, 2018) and to be suitable for morphological post-examination or DNA re-extraction for DNA barcoding of individual specimens (Batovska et al., 2021). Furthermore, communities recovered from non-destructive DNA extraction alternative treatment have been shown to be more similar to homogenate treatment than what collection medium metabarcoding detected, respectively (Kirse et al., 2022). However, non-destructive DNA extraction was found to be sometimes partially destructive after a long incubation time (e.g., overnight lysis), especially for soft-bodied taxa like Diptera (Marquina et al., 2022; Kirse et al., 2022). A recent study did report a successful attempt—albeit limited to controlled and species-poor community samples—of non-destructive DNA extraction from a mix of extraction buffer and propylene glycol acting as preservative solution (Martoni et al., 2021). Nevertheless, these non-destructive alternatives may be limited in terms of scalability by the important volumes and associated costs of manufactured extraction buffer required, ranging from 55–65 U.S. $ per Malaise trap sample (Kirse et al., 2022). Although homemade extraction buffer can drastically reduce costs nearly ten-folds (Marquina et al., 2022), this option may reduce cross-comparability between studies and laboratories. Finally, to aim at recovering abundance regardless of the sample treatment method, optional and additional molecular steps such as DNA spike-in of known mock communities and DNA concentration can be implemented to infer the relative abundance of taxa from sequence read-based number correction (Luo et al., 2022). Still, this method would gain further tests on non-destructive methods, with post-control and validation after secondary morphological sorting of the intact vouchers.

Conclusion

We evaluated the feasibility of detecting terrestrial arthropod species from collection media of two mass-trapping systems using metabarcoding. We tested metabarcoding of monopropylene glycol (MPG)-water collection medium from window-flight traps for the first time, but show that it failed to recover significant numbers of species. In contrast, metabarcoding of MPG-ethanol collection medium from Malaise traps showed a high species recovery and higher number of small and soft-bodied arthropod species compared to destructive metabarcoding of homogenates. However, methodological choices prevent us from making meaningful comparisons of the invertebrate communities found for each treatment, and highlight the need for further protocol optimization. Indeed, analyzing the collection/preservation medium takes metabarcoding away from ideal experimental conditions and we call for caution in considering the potentially significant impact of fieldwork conditions (DNA degradation, inhibitors, collection medium composition), laboratory processes (DNA filtering and extraction methods, primer affinity) and data analysis (sequence length, sequencing depth) on the results. Nevertheless, further developments may eventually unlock the full potential of this approach—a goal worth pursuing, especially when working on biodiversity hotspots where many specimens collected are new to science (Lopez-Vaamonde et al., 2019), and must be therefore preserved.

Portions of this article were previously published as part of a preprint (https://doi.org/10.1101/2023.02.07.527242).

Supplemental Information

Supplemental Information 1 Supplementary tables

Click here for additional data file.

Supplemental Information 2 Sequencing success for collection medium treatment

Click here for additional data file.

Supplemental Information 3 MOTU and read numbers after filtering steps of Malaise trap datasets generated with different bioinformatic demultiplexing thresholds

Circles represent the number of MOTUs retained for various filtering and demultiplexing stringency thresholds, with circle wideness corresponding to the associated read numbers. Bioinformatic combination parameters are defined by the number of PCR replicates in which a MOTU with a minimum of two reads has to appear to be retained (i.e., MOTU present with two reads in at least 1/6 PCR, overlapping 2/6, 3/6 or 4/6 PCR replicates, coloured from lighter to darker yellow, respectively). Filtering steps are described as follow: Raw correspond to the dataset recovered after demultiplexing; Arthropod only indicates a filtering based on taxonomy to retained MOTUs identified as Arthropods only; Similarity ¿80% corresponds to a filtering based on the percentage of similarity to arthropod sequences shared with the consensus from BOLD database used for taxonomic identification and keeping MOTUs sharing at least 80% similarity only; MT filtered corresponds to the final dataset used for Malaise traps, with a merging of MOTU and occurrence information based on an identical species identification.

Click here for additional data file.

Supplemental Information 4 MOTU richness for different taxa recovered from Malaise traps using metabarcoding of collection medium (Coll. Med.) or homogenate (Hom.) with the Uni-Minibar (U-M) or Leray/Geller (L/G) primer sets

Boxplot of MOTU count for collection medium (yellow) or homogenate metabarcoding (blue) with Uni-Minibar primer set or from homogenate metabarcoding using mlCOIintF/jgHCO2198 primer set (grey) of the same Malaise trap samples. Black dots represent samples considered after bioinformatic processing and data curation. Significant differences adjusted with Bonferroni correction are highlighted with ‘*’ and ‘N.S.’ stands as non-significant. Studied taxa are: (A) non Insecta (i.e., Arachnida and Collembola) (Pairwise T-test: 1–2: p = 6.6e−03; 1–3: p = 7.4e−05; 2–3: p = 1); (B) Diptera (Wilcoxon rank sum-test: 1–2: p = 0.15; 1–3: p = 6.0e−09; 2–3: p = 8.3e−05); (C) Hymenoptera (W-test: 1–2: p = 1.9e−03; 1–3: p = 7.6e−10; 2–3: p = 1.2e−05); (D) Coleoptera (W-test: 1–2: p = 3.9e−03; 1–3: p = 6.6e−09; 2–3: p = 4.2e−03); (E) Lepidoptera (W-test: 1–2: p = 1.4e−02; 1–3: p = 1.5e−09; 2–3: p = 3.1e−05); (F) other Insecta orders grouped (W-test: 1–2: p = 1; 1–3: p = 7.7e−08; 2–3: p = 3.5e−04).

Click here for additional data file.

We are thankful to Wilfried Heintz, Laurent Burnel, Jérôme Molina and Jérôme Willm for the field work, Sylvie Ladet for her SIG assistance, Carl Moliard and Guilhem Parmain for the sorting and morphological identification of the Coleoptera from WFTs. We also want to thank Florent Figon and Sophie Van Meyel for helpful discussions on the manuscript. Part of this work has been carried out with the technical support of the Genomic Facilities (PST ‘Analyse des Systèmes Biologiques’) at the University of Tours. We are thankful to Dr. Caroline Chimeno and an anomynous reviewer for their insightful comments that greatly improved the manuscript.

Additional Information and Declarations

Competing Interests

Author Contributions

Field Study Permissions

Data Availability

The authors declare there are no competing interests.

Lucas Sire conceived and designed the experiments, performed the experiments, analyzed the data, prepared figures and/or tables, authored or reviewed drafts of the article, and approved the final draft.

Paul Schmidt Yáñez analyzed the data, authored or reviewed drafts of the article, and approved the final draft.

Annie Bézier performed the experiments, authored or reviewed drafts of the article, and approved the final draft.

Béatrice Courtial performed the experiments, authored or reviewed drafts of the article, and approved the final draft.

Susan Mbedi performed the experiments, authored or reviewed drafts of the article, and approved the final draft.

Sarah Sparmann performed the experiments, authored or reviewed drafts of the article, and approved the final draft.

Laurent Larrieu conceived and designed the experiments, authored or reviewed drafts of the article, and approved the final draft.

Rodolphe Rougerie conceived and designed the experiments, authored or reviewed drafts of the article, and approved the final draft.

Christophe Bouget conceived and designed the experiments, authored or reviewed drafts of the article, and approved the final draft.

Michael T. Monaghan conceived and designed the experiments, authored or reviewed drafts of the article, and approved the final draft.

Elisabeth A. Herniou conceived and designed the experiments, authored or reviewed drafts of the article, and approved the final draft.

Carlos Lopez-Vaamonde conceived and designed the experiments, authored or reviewed drafts of the article, and approved the final draft.

The following information was supplied relating to field study approvals (i.e., approving body and any reference numbers):

No ethical approval was required for the sampling of insects as study organisms.

The following information was supplied regarding data availability:

All supplementary information, analytical script and sequencing data are available at Figshare: Sire, Lucas (2023). Supplemental Data - Non-destructive DNA metabarcoding of arthropods using collection medium from passive traps. figshare. Dataset. https://doi.org/10.6084/m9.figshare.22094036.v4.

The raw sequencing data are also available at NCBI CLIMTREE_EtOH: PRJNA927244.

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
