# Peer review of "Persisting roadblocks in arthropod monitoring using non-destructive metabarcoding from collection media of passive traps"

_PeerJ, doi:10.7717/peerj.16022_

## Round 0.1 · original submission · Major Revisions

Dear Dr. Sire,

Thanks for your interesting manuscript sent to Peer J. Although the manuscript is interesting and important, Reviewer 1 pointed out important issues that should be addressed before the manuscript is considered for publication. The reviewer also provided the Word file which should help you to further improve your text and put the manuscript in a broader context for a reader without experience in the area. Reviewer 2 (pdf attached) also provided a very thoughtful review (four pages) and points out the importance and particularly that the article should be reframed as “what not to do”. Both reviewers also pointed out missing or improper references so please address this issue as well. I ask you to do a point-by-point reply to the reviewers.

·

Basic reporting

I was able to access all data that was mentioned in the data deposition statement. Well done. I suggest that the authors also upload their R script to Figshare, as it allows the reproducibility of results, helps fellow scientists in their struggle in using R, but also permits review of the data.

Overall, this manuscript complies with baseline criteria. It is well-set against the scientific background. It addresses a very important topic regarding the use of different DNA sources in metabarcoding approaches. Reading the manuscript, it is clear that the authors have put a lot of effort into their study. The Introduction reads very well and provides a great overview of the current problematic while emphasizing the importance of the study.

However, I found both the Methods and Results sections rather difficult to read and would highly recommend the authors to:

- Make these sections as concise as possible. I think a lot of the data (e.g. PCR conditions) can be summarized in a table. I know that it is difficult to keep the Methods concise, however, because the authors are working with four different datasets (MT homogenates, MT collection medium, WFT morphological, and WFT collection medium), I recommend using more subsections to facilitate reading. I had to read some passages more than once to clearly understand what was done with each dataset, so I assume readers with less knowledge in this field of science might have trouble following the script.

- Polish the language. The Methods and Results have been clearly written by a separate person than the Introduction section – this is of course fine, however, I would ask the authors to see that the style of writing is more even between sections. Just a personal preference, but I am a fan of active voice regarding the Methods. YOU conducted the sampling, YOU went into the laboratory, then why not write “We performed a PCR on…”? It gives the readers an automatic link to the authors, and it is clear that all of this was conducted by the authors especially as today, a lot of lab work is done externally.

Experimental design

This study is original primary research, and as mentioned above, addresses a prevailing research question. The aims of the study are well-defined and very meaningful. The experimental design is of high quality and produces useful data. Due to laboratory issues regarding the amplification of the 313-bp fragment, the authors had to resort to amplifying much shorter COI fragments, however, I find it commendable how the authors have dealt with the problematic of using these too short (and non-overlapping fragments) when comparing across methods. Well done.

As mentioned above, I find it necessary for the authors to restructure their Methods to make reading more intuitive. Instead of describing how steps were different across the datasets, I would recommend creating a subsection for each dataset, including the processing steps from start to finish.

Validity of the findings

I find that the findings (that metabarcoding of the collection fluid can be used as a complement to homogenate sequencing) of this research need to put into better context, as they are in congruence with some of the previous works. I also find it important for the authors to separate their results based on the methodology (see geeral comments). Based on the results, I argue that sequencing the collection fluid of WFTs is not a method that produces valid results, especially because in 30% of samples, absolutely no MOTUs were recovered, and overall they found very few species (3 of Coleoptera).

Additional comments

Please also refer to the Microsoft Word Document where I have added comments.

- Figure 4: I believe this figure could be drastically simplified. I personally would remove all columns of order, family and genus because overall, what’s important is how many MOTUs you recovered per order, how many of these were identifiable to the species-level. Because you are comparing between both approaches, it would be best to merge these results (as in C), however, use stacked columns. For example, homogenates underneath, overlap, and collection fluid above (see Chimeno et al., 2022b).

- lines 376-393: This passage is very tedious to read. You can simplify this by only taking the more taxa into account, or structuring this passage better.

- Regarding demultiplexing – being that I am not an expert in this field, I had a few difficulties understanding this section. I would therefore simplify everything as best as possible. A lot of these results can be displayed in a table.

- Also, instead of using “/ ” to address proportions, I recommend using percentages. So instead of (see line 297) “reads were found in 2/11 negative controls…” I would opt for “We found reads in 18% of our negative controls…” (see what I did there with the active voice - sneaky sneaky). Or instead of (see line 295-296) “…from additive (including MOTUs present in at least one of the total six PCR replicates) to more stringent demultiplexing threshold (including MOTUs that are present in at least two, three, and four of the six PCR replicates).

- Note that numbers 1-10 are always written out.

- I recommend reducing the number of Figures. For example, Fig. 4, Fig. 5 and Fig. 6 can be summarized into a single one.

- The authors need to better incorporate the finding of previous works into their discussion. For example, in line 505, the authors mention that they are surprised by the fact that they have recovered very different communities with each approach. This is not surprising, as this has been already discussed several times in current literature. There are many reasons why this is the case, and I think it is crucial that the authors address these too. This also applies to lines 531-537. Here the authors do not discuss the differences between metabarcoding of terrestrial and aquatic invertebrates, although it is well-researched why metabarcoding the fluid for aquatic samples is much more successful than terrestrial.

- Lines 511-515: This is a weird sentence, please rephrase.

- Lines 515-519: I don’t understand the link between dipterans being important and ETOH-MPG medium metabarcoding. Please elaborate.

- Lines 538-540: Why would these have comparable value if you obtain different communities?

- One aspect that is lacking in this manuscript is the mentioning and discussion of non-destructive DNA extraction methods that are currently flourishing.

- The discussion needs better structuring as they are working with many different datasets, the authors need to clearly state their points in regard to each.

Reviewer 2 ·

Basic reporting

Basic reporting correct.

Experimental design

See attached document.

Validity of the findings

See attached document.

Annotated reviews are not available for download in order to protect the identity of reviewers who chose to remain anonymous.

---

## Round 0.2 · Minor Revisions

Dear Dr. Sire,

Attached the comments of the single reviewer, the manuscript has been largely improved.

Although minor, please address all the comments of this reviewer before the paper can be considered for publication.

·

Basic reporting

The authors have done a formidable job in implementing both reviewers’ comments, and it is obvious that they put much effort and time into doing so. The manuscript is more concise, reads well, overall very improved. I just have minor suggestions (see below). The discussion reads well and addressed all important point. Recent literature has been added. Well done. Also, the figures are great.

Experimental design

The authors clearly cite their research goal. The methods have become more concise and easier to understand, and the figures have been changed accordingly.

Validity of the findings

Nothing to report - the authors have attached all necessary data.

Additional comments

line 134: add „in“ the center…

line 139: remove “containing the collection medium as well as the arthropods“

line143: add “the” arthropods from “the” WFT collection media … filtered them from “the” MT … using “a” single-use..

line153: add to “a” fine powder

line 156: change sentence to: we performed DNA extraction on 25 mg (…) of the arthropod powder using the Qiagen ….

line188: from which traps?

line255 : add “the” BOLD reference database

line280: “see” written small

I would simplify and also structure this differently, for example:

Processing the window-flight traps (WFTs) using 1-4/6 replicate combination parameters, collection medium sequencing yielded 191, 77, 53, and 37 MOTUs respectively. In all cases, more than half of MOTUs were represented by Diptera, ranging from 52-56%.

You do not need to write in parentheses how many MOTUs you obtained each time, you can just add them to a table. Also, I did say to always write out numbers below 10, however, in such cases with listing numbers, you do not need to do this.

line 283: However, only 4-10% (with a total of 20, three, two and two MOTUs respectively) of Coleoptera MOTUs were recovered, albeit being the main taxonomic group sampled by WFT.

This sentence, I would move to the discussion, but remove (with a total of 20, three, two and two MOTUs respectively), this you can show in a table.

line 288: add to “the” species level

line 301: remove “s” in combinations

line 302 replace “drastic loss” with “drastic decrease” in both read- and MOTU- numbers

line 349: add “the” WFT collection…
also, remove “only (twice)”. In the results, just present what you got without putting value to it. You can discuss this in the discussion.

lines 355-357: Please rephrase, this sentence is weird to understand.

line 357: add “the” EtOH-MPG …

line 368: remove “s” in MOTUs

---

## Round 0.3 · accepted · Accept

Congratulations. All the required changes were performed and the manuscript is accepted;